# Experimental Microwave Complex Conductivity Extraction of Vertically Aligned MWCNT Bundles for Microwave Subwavelength Antenna Design

**DOI:** 10.3390/mi10090566

**Published:** 2019-08-27

**Authors:** Charlotte Tripon-Canseliet, Stephane Xavier, Yifeng Fu, Jean-Paul Martinaud, Afshin Ziaei, Jean Chazelas

**Affiliations:** 1Physics and Material Science Laboratory, Sorbonne Universités, CNRS-ESPCI, 75005 Paris, France; 2THALES Research and Technology, 91767 Palaiseau, France; 3Electronics Materials and Systems Laboratory, Department of Microtechnology and Nanoscience, Chalmers University of Technology, SE - 412 96 Gothenburg, Sweden; 4THALES Defence Mission Systems, 78 851 Elancourt, France

**Keywords:** multi-wall carbon nanotubes, microwave impedance, small antennas

## Abstract

This paper reports the extraction of electrical impedance at microwave frequencies of vertically aligned multi-wall carbon nanotubes (VA MWCNT) bundles/forests grown on a silicon substrate. Dedicated resonating devices were designed for antenna application, operating around 10 GHz and benefiting from natural inductive/capacitive behavior or complex conductivity in the microwave domain. As obtained from S-parameters measurements, the capacitive and inductive behaviors of VA MWCNT bundles were deduced from device frequency resonance shift.

## 1. Introduction

Carbon nanotubes (CNTs) have been extensively studied over the last decades due to their exceptional electrical, thermal, and mechanical properties. In addition to these properties, applications of CNTs in the microwave domain have also appeared, as numerous research works have been devoted to the elaboration of active and passive radio-frequency (RF) devices such as resonators and field-effect transistors (FETs) [1,2], but also to the development of chemical and mechanical sensors [3,4,5] for environment monitoring and biomedical applications. Several technical approaches are competing for microwave antenna miniaturization prospects, as the resolution of numerous technological processes is converging drastically down to the nanometer scale. At the same time, hybrid material compatibility can now be implemented as high-resolution characterization tools such as near-field probe techniques coupled to large broadband vector network analysis become more affordable [6,7,8,9]. 

Major challenges remain in the qualification of this material in the microwave domain in terms of complex impedance or conductivity. In this research field, 1D and 2D materials have recently validated their eligibility of implementation in microwave circuit design, reaching a high level of innovation, as confirmed from theory [10,11]. Indeed, graphene and metallic CNT material conductivity, offering a non-negligible imaginary part, now stands as the best candidate for antenna miniaturization (Table 1) following the initial experimental validations [12]. These technical approaches are competing with surface-mode propagation solutions superimposed by metallic/dielectric interfaces. Plasmonic structures are the most studied nanoscale configuration in the THz regime [13].

At microwave frequencies, metasurface circular topologies leading to a leaky-wave radiation such as in [14,15] allow a maximum miniaturization factor of λ/50 for each individual element. CNT-based composite materials technology in which multi-wall (MW) or single-wall (SW) CNTs are dispersed randomly in a solution or polymeric layer have demonstrated their efficiency in low-frequency antenna design on flexible substrates [16].

At present, the identification of electromagnetic properties at RF/microwave frequencies has become a milestone in order to understand and design new components for future implementation in the next generation of CNT-based microwave systems. 

In this paper, we report on the electromagnetic material properties of vertically aligned CNT bundles grown on a substrate by the exploitation of a de-embedding technique developed for integrated technology. Using this approach, the experimental complex conductivity of MWCNTs bundles processed from well-controlled vertical growth technology in the microwave domain is presented from impedance extraction obtained from on-wafer S-parameters measurements.

## 2. MWCNT Material Properties from Theory 

An individual single-wall carbon nanotube (SWCNT) consists of a spatially unique angled rolling of a graphene sheet that depends on the associated chiral vector, which defines its degree of metallicity. 

Frequency-dependent graphene conductivity can be approximated using the Kubo formula (1), with the chemical potential μ_c_ and the relaxation rate Γ. This complex expression, in which Γ was experimentally estimated to be 0.1 meV, exhibits a complex behavior with σ_real_ and σ_imag_ as real and imaginary parts of σ_s_(ω) depending on the value of μ_c_. From a dyadic Green’s function formulation of electromagnetic field propagation through Sommerfeld integrals, Transverse Electric (TE) or Transverse Magnetic (TM)-mode surface wave propagation operations have been validated for positive or negative values of σ_imag_ from [3].

Starting from the graphene conductivity formulation (1), a specific SW rolling configuration leading to a metallic behavior also implies a complex conductivity definition as in a chiral case (2). A MW formation of CNTs has to be selected in order to ensure a metallic behavior from catalyst atom choice and growth technique. From the literature, an RLC circuit representation study of individual SW or MW CNTs from electron gas theory states a high linear resistance, static capacitance, and kinetic inductance per micrometer values at several orders of magnitude higher than a standard metallic gold wire with the same dimensions (i.e., with 5 nm radius and 300 µm distance from a ground plane; Table 1).
(1)$$\sigma_{s}\left(\omega \right)=i\frac{1}{\pi \hbar^{2}}\;\frac{e^{2k_{B}}T}{\omega +i2\Gamma }\;\left\{\frac{\mu_{c}}{k_{B}T}+2\;\ln\left[\exp\left(-\frac{\mu_{c}}{k_{B}T}\right)+1\right]\right\}+i\frac{e^{²}}{4\pi \hbar }\ln\left[\frac{2\left|\mu_{c}\right|-\hbar \left(\omega +i2\Gamma \right)}{2\left|\mu_{c}\right|+\hbar \left(\omega +i2\Gamma \right)}\right]$$
(2)$$\sigma_{zz_{\mathrm{chiral}}}=-j\frac{8\pi e^{2}\gamma_{0}\sqrt{3}}{h^{2}\sqrt{m^{2}+n^{2}\left(\omega -j\nu \right)+mn}}\;\mathrm{with}\;2n+m=N$$
(3)$$\sigma_{\mathrm{bundle}}=\sigma_{b0}+j\sigma_{b}\left(\omega \right)$$

As a consequence, from these statements, the existence of an imaginary character of electrical conductivity (i.e., an inductive or capacitive behavior in frequency) presents strong interest in electronic circuits designed from the material itself. Furthermore, considering a vertically aligned collective representation of individual CNTs as in a bundle, a complex expression of bundle-equivalent conductivity must be assumed as in (3), with a static parameter σ_b0_ and a frequency-dependent imaginary part σ_b_(ω).

## 3. Microwave Material Parameters Identification Procedure

### 3.1. Microwave Structures Design

Preliminary resonant structures in coplanar waveguide (CPW) technology with suitable taper-based electrodes (Figure 1) were designed using commercial 3D electromagnetic simulation code (Ansys-HFSS) with technological process implementation.

For characterization technique improvements, a new differential-type approach is chosen by assuming a dedicated de-embedding structures definition. In order to overcome the physical constraints imposed by the material dimensions and the technological process, a transmission line circuit topology in CPW technology was selected, with a tapered-type profile which preserves electrical contact from standard coplanar access to the circular cross section of the MWCNT bundle.

### 3.2. Material Microwave Conductivity Extraction Procedure

From material considerations, insertion of the MWCNT bundle at the end of the electrode adds a complex impedance in series, which contributes to a resonant frequency shift as a reactance modification of the system. By a classical spatial integration of the bundle conductivity from (2) involving a bundle length *L* and diameter *D*, an equivalent impedance *Z_bundle_* approach becomes a comprehensive representation of the added material at mesoscopic scale (4).

By introducing (2) in (4) and assuming that *Z_bundle_* = *Z*′*_bundle_* + *j Z*″*_bundle_* with a form factor *F* = *L*/*D*, real (*Z*′*_bundle_*) and imaginary (*Z*″*_bundle_*) parts of *Z_bundle_* relations become inversely frequency dependent, as expressed in (5) and (6):(4)$$Z_{bundle}=\frac{1}{\sigma_{bundle}}\cdot \frac{4L}{\pi D^{2}}$$

(5)$${Z^{\prime }}_{bundle}=\frac{\sigma_{b0}}{\sigma_{b0}^{2}+\sigma_{b}{\left(\omega \right)}^{2}}\cdot \frac{4\pi F}{D}$$

(6)$$Z{\prime \prime }_{bundle}=-\frac{\sigma_{b}\left(\omega \right)}{\sigma_{b0}^{2}+\sigma_{b}{\left(\omega \right)}^{2}}\cdot \frac{4\pi F}{D}.$$

## 4. Experimental Results

### 4.1. Technological Process

The vertical technology developed for device elaboration relies on a multistep process. In a first step, a 2-µm-thick layer of Mo was sputtered on a highly resistive (HR) Si/SiO_2_ substrate for the definition of microwave structures. After optical lithography and ion beam etching (IBE), Al_2_O_3_ and Fe layers were locally deposited as catalyst for the thermal Chemical Vapor Deposition (TCVD) growth of MWCNT bundles at 700 °C, allowing for the growth of a bundle with a form factor *F* equal to 5.

### 4.2. Experimental Microwave Environment

A broadband microwave experimental setup based on a probe test equipment connected to a vectorial network analyzer was used for the on-wafer S-parameters measurements of devices after an on-wafer Short/Open/Load/Thru (SOLT) calibration on alumina substrate in the 0.2–67 GHz frequency band.

For microwave signal coupling methodology, different device inner electrodes (C1 to C5) were designed and tested in order to validate the material properties. Each design was replicated on the same wafer as two sets of samples were processed with and without CNT bundles on the same wafer, maintaining the same bundle diameter of 20 µm for each device, in order to extract the VA (vertically aligned) MWCNT bundle impedance contribution from the microwave test structure itself. 

From reflection coefficient measurements, microwave input impedances *Z_in_* and *Z*′*_in_* were extracted from devices with and without VA MWCNT bundle implementation, as in Figure 2. As VA MWCNT bundles are electrically connected in series to microwave coplanar structures, the frequency-dependent complex impedance of VA MWCNT bundles was extracted from the reflection coefficient to input impedance conversion followed by a de-embedding technique assumed by direct impedance subtraction.

As expected from theory and as shown in Figure 3a, a decrease of the real part of the impedance in frequency was confirmed from five samples measurements. At low frequencies, an impedance value of 80–100 Ω validates the collective response of the individual shunt high resistance of each MWCNT forming the bundle, with a density of 10^15^ units per cm^2^. At higher frequencies, VA MWCNT bundles demonstrated a decreasing frequency-dependent impedance as expected from Equations (5) and (6), as well as a capacitive behavior from its negative imaginary part. 

In addition, a non-negligible and complex conductivity in Figure 4 attributed to this material was experimentally observed from the five samples for the first time at microwave frequencies. These exceptional properties confirm this new material’s implementation as a disruptive technology for the next generation of microwave devices designed with a high degree of miniaturization.

## 5. Conclusions

Preliminary research works were performed on sub-wavelength MWCNT-based antennas designed benefiting from the natural inductive/capacitive behavior or complex conductivity never before achieved in classical conductors in the microwave domain. By also exploiting a vertically aligned CNT bundle configuration that drastically reduces the contact resistance of individual MWCNTs, the electromagnetic VA MWCNT bundle properties as obtained by equivalent complex impedance extraction from our experimental material process were identified for the first time using CPW technology, from 5 to 15 GHz frequency. These exceptional properties concretize the eligibility of this new material as a disruptive technology for the next generation of microwave devices and antennas, at the material level. Future works will focus on the determination of VA MWCNT bundle complex impedance law in respect of physical dimensions and cross section profile.

## 6. Patents

This work has led to the filing of a patent under deposition number FR1800496. 

## Figures and Tables

**Figure 1 micromachines-10-00566-f001:**
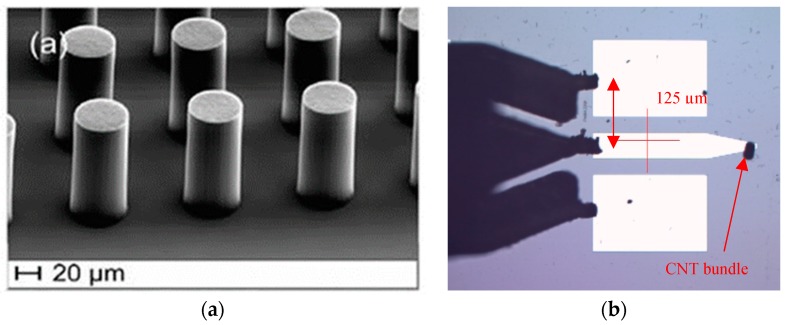
Technological process and Characterization of the vertically aligned multi-wall carbon nanotubes (MWCNTs)-based device. (**a**) SEM (scanning electron microscope) image of CNT bundles grown by Thermal Chemical Vapor Deposition (TCVD) (**b**) Top view of MWCNTs-based device in CPW (coplanar waveguide) technology under a 125-µm-pitch test probe (optical microscope image).

**Figure 2 micromachines-10-00566-f002:**
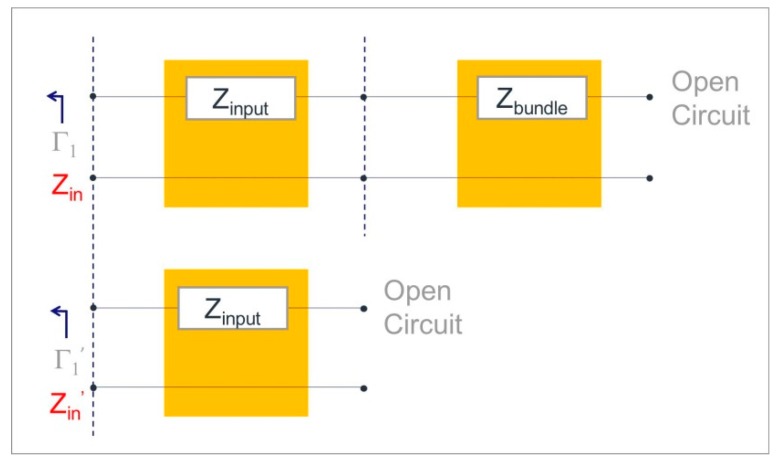
Schematic representation of the complex impedance extraction of VA (vertically aligned) MWCNT bundles from a two-device set.

**Figure 3 micromachines-10-00566-f003:**
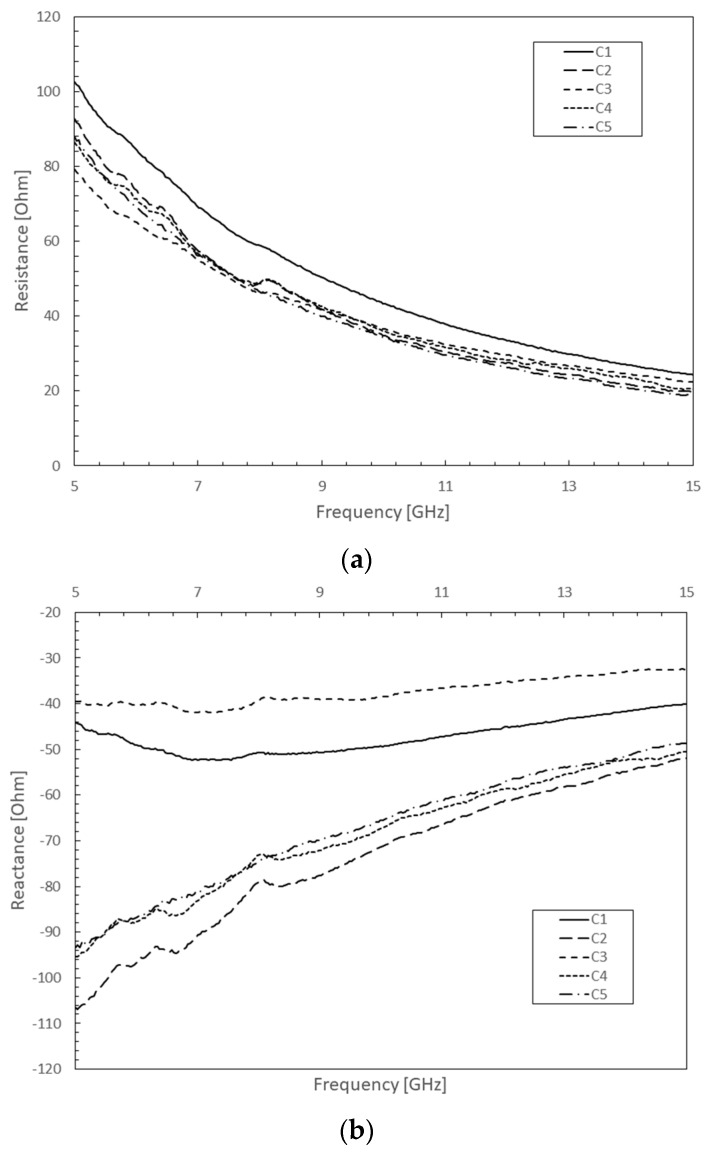
Experimental VA MWCNT bundle impedance extracted from five different CPW devices (C1 to C5) incorporating identical bundle dimensions in the 5–15 GHz frequency band: (**a**) Real part; (**b**) Imaginary part.

**Figure 4 micromachines-10-00566-f004:**
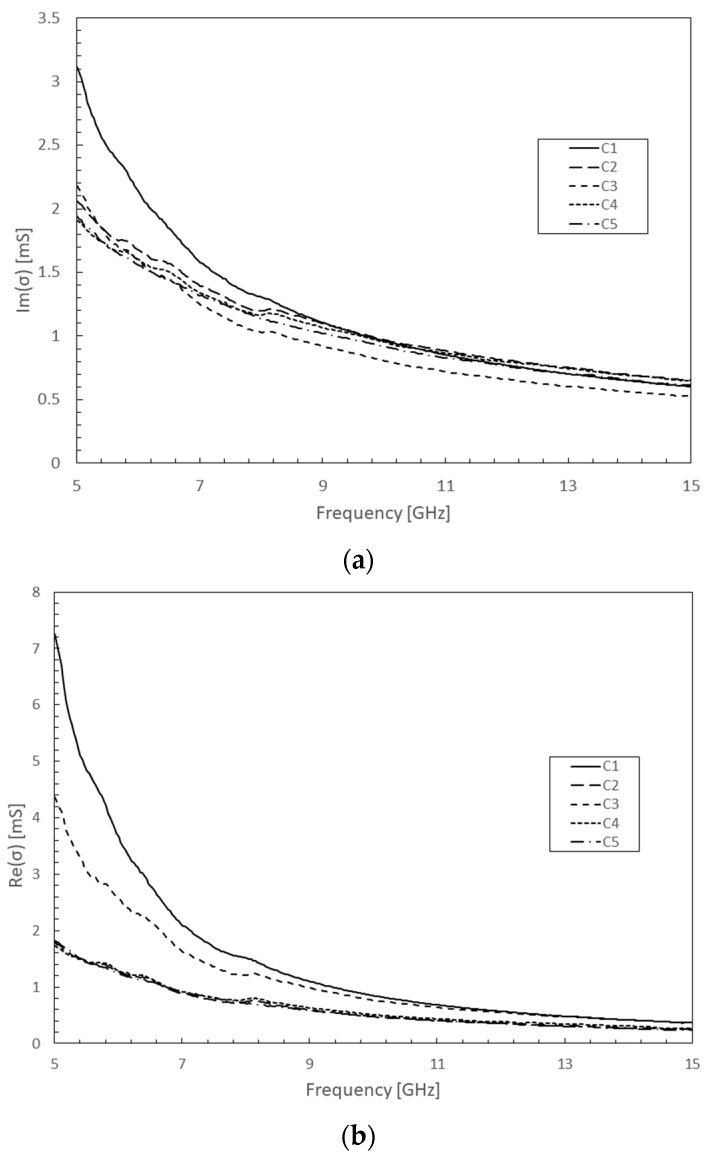
Experimentally extracted VA MWCNT complex conductivity measurements from five microwave devices (C1 to C5) incorporating identical bundle dimensions: (**a**) Real part; (**b**) Imaginary part.

**Table 1 micromachines-10-00566-t001:** Comparison of lineic electrical properties of a 10-nm-diameter cylinder.

Parameter	Perfect Electrical Conductor (PEC) Wire	Individual MWCNT (Multi-Wall Carbon Nanotube)
Resistance	5 mΩ/µm	7 kΩ/µm
Inductance	1 pH/µm	20 nH/µm
Capacitance	5 aF/µm	0.5 fF/µm

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
