# Peer review of "Experimental Microwave Complex Conductivity Extraction of Vertically Aligned MWCNT Bundles for Microwave Subwavelength Antenna Design"

_micromachines, 2019, doi:10.3390/mi10090566_

Round 1

Reviewer 1 Report

Although applications of CNTs  into microwave antenna may be interesting in the electromagnetic industry, we cannot evaluate the applicability expressed  by the authors because lack of experimental details such as the five microwave devices (C1 to C5) having different inner electrodes, experimental conditions of the growth procedures  of CNT, and microscopic structures of MWCNT. I recommend that the authors adds the experimental information into the original manuscript for the potential readers and  researchers who want to use this kind of antenna devices.

Author Response

Dear Reviewer #1,

Thank you very much for your constructive comments and remarks.

Indeed some strong revisions were required. We added some more information on our measurement procedure which was not so clear indeed.

We tried also to improve English even if it is not our mother language. 

Please find the revised version, where modifications have been highlighted in red.

Kind regards,

C. Tripon-Canseliet

Reviewer 2 Report

Dear authors,

Thank you for your manuscript. Could you please address following points:

- improve the quality of English,

- usage of abbreviations, Figures (and their quality) and corresponding numbering, numbers and units, subscripts and equations, reference styling in the manuscript,

- improve the state of the art summary and in general introduction,

- methodology - which de-embedding structures did you use, detailed approach on how did you extract impedance of your structures, schematic models and corresponding correlation to manufactured devices, and so on...,

- more detailed discussion of results and variation between results,

- in general, the reading flow of the manuscript should be improved and please remove descriptions of sections from the final manuscript (i.e. line 87-89, 143-145),

- table 1 - avoid wording "few" - we are dealing here with an exact science. Thus, adjust your table accordingly,

- could you please change wording Fig. 4a, or add Fig. 4a if needed.

Author Response

Dear Reviewer #2,

Thank you very much for your constructive comments and remarks.

Indeed some strong revisions were required.

We have made revisions on each equests as:

improve the quality of English: Done

1) usage of abbreviations, Figures (and their quality) and corresponding numbering, numbers and units, subscripts and equations, reference styling in the manuscript; done

2) improve the state of the art summary and in general introduction: done

3) methodology - which de-embedding structures did you use, detailed approach on how did you extract impedance of your structures, schematic models and corresponding correlation to manufactured devices, and so on...,: done

4) more detailed discussion of results and variation between results: done

5) in general, the reading flow of the manuscript should be improved and please remove descriptions of sections from the final manuscript (i.e. line 87-89, 143-145): done

6) table 1 - avoid wording "few" - we are dealing here with an exact science. Thus, adjust your table accordingly: done

7) could you please change wording Fig. 4a, or add Fig. 4a if needed: done

Please find the revised version, where modifications have been highlighted in red.

Kind regards,

C. Tripon-Canseliet

Round 2

Reviewer 1 Report

Unfortunately, I could not understand the difference in dimension among the tested internal electrodes (C1 to C5) and dimension of CNT arrays (length (500 um ?), diameter, density, and number of walls) , all of which I have recommended that the author make clear. Although the effect of CNT on the performance of microwave detection may be clearly detected by the author, we cannot reproduce similar experiments because of the lack of experimental requirements. If the author cannot make them clear for commercial contracts, I recommend that the author explain the mechanism of the improvement for antennas by adapting VACNT having typical dimensions.

Author Response

Unfortunately, I could not understand the difference in dimension among the tested internal electrodes (C1 to C5) and dimension of CNT arrays (length (500 um ?), diameter, density, and number of walls) , all of which I have recommended that the author make clear.

The form factor of 5 was applied on a CNT bundle of 20 µm in diameter, leading to a length of 100µm.

The approximated CNT density from SEM image leads to a value of 1011/cm². TEM image states a number of walls equal of 2-3.

Although the effect of CNT on the performance of microwave detection may be clearly detected by the author, we cannot reproduce similar experiments because of the lack of experimental requirements. If the author cannot make them clear for commercial contracts, I recommend that the author explain the mechanism of the improvement for antennas by adapting VACNT having typical dimensions.

Yes, you are right, we are restricted to confidentiality from active industrial contracts. Some corrections have been done in the nex revised version. Please refer to the revised text as these requests were also sent by the second reviewer.

Reviewer 2 Report

Dear authors,

Please find comments below.

line 18: electrical impedance at microwave frequencies

line 29/30: please check wording in the sentence (verbs)

line 30/31: references to applications?

line 32/33: -||-

line 38: what do you mean by impedance/conductivity? is it one, or another?

line 57: in this paper, we report on the development of a de-embedding 

technique dedicated vertically aligned CNT.

line 76/76: spacing between units and numbers missing

line 97: could you please indicate antenna elements on Fig. 1. I see a short

 cpw transmission line with a taper loaded with nanotubess (i presume). Coud you also

add dimensions on Fig 1b. Finally, could you please state in your opinion how is signal

connected to the ground after reaching cnts (what is the return path for the signal?)

line 101: what is in your case dedicated de-embedding structure definition?

line 137: could you please add pictures of manufactured devices

line 143: which de-embedding technique? Additionally, to be able to substract impedances

readers would agree with this to know the lengths of your devices?

line 152: speaking of antenna? Could you please define your antenna in the paper?

Is is the complete structure or only cnts? Did you check how your antenna performs?

line 169: could you please indicate which data has been reveled at 60 GHz? I am

having difficulty finding it.

line 171: TE/TM in progress. You have not talked about it in the paper, and you 

have stated in the abstract that this behaviour was analyzed.

Author Response

Reviewer #2

line 18: electrical impedance at microwave frequencies- Done in the text

line 29/30: please check wording in the sentence (verbs) – Done in the text

line 30/31: references to applications - done

line 32/33: -||-Revision done in the text

eline 38: what do you mean by impedance/conductivity? is it one, or another? Done in the text

line 57: in this paper, we report on the development of a de-embedding technique dedicated vertically aligned CNT. - Done in the text

line 76/76: spacing between units and numbers missing - Done in the text

line 97: could you please indicate antenna elements on Fig. 1. I see a short

 cpw transmission line with a taper loaded with nanotubess (i presume). Coud you also

add dimensions on Fig 1b - Done in the text

Finally, could you please state in your opinion how is signal connected to the ground after reaching cnts (what is the return path for the signal?) The GSG probe ground is connected to the chuck. So at the CNT bundle location the E-field msut be parallel to the bundle

line 101: what is in your case dedicated de-embedding structure definition? It involves a sum of impedances in series which can be extracted by subtraction if one of this impedance is eliminated

line 137: could you please add pictures of manufactured devices. We cannot for active industrial contracts

line 143: which de-embedding technique? From S11 parameter, we can calculate the device input impedance of each device defined by  a sum of equal impedances. Five sets of two identical CPW devices in all dimensions (one contains CNT bundle, this other one not) are processed and tested on the same wafer. Additionally, to be able to substract impedances readers would agree with this to know the lengths of your devices?

line 152: speaking of antenna? Could you please define your antenna in the paper? There is no antenna at this step (mistake made in the title of the figure ) , only a resonant device

Is is the complete structure or only cnts? Only CNT bundle after impedance de-embedding

Did you check how your antenna performs? NA then

line 169: could you please indicate which data has been reveled at 60 GHz? I am

having difficulty finding it. You are right, we have measurements but not in this draft paper. Text changed then

line 171: TE/TM in progress. You have not talked about it in the paper, and you 

have stated in the abstract that this behaviour was analyzed. You are also right, we have tested samples with different E-field orientation but not promoted in this this draft paper. Text changed consequently.

Round 3

Reviewer 1 Report

I recommend that this work be published not as a regular paper but as a technical note, because the experimental details have not been explained yet. for example, details of the used device inner electrodes (C1 to C5).